# Flowering agricultural landscapes enhance parasitoid biological control to *Bemisia tabaci* on tomato in south China

Shaowu Yang[1,2], Wenjun Dou[1,3], Mingjiang Li[1], Ziliao Wang[1], Guohua Chen[1]*, Xiaoming Zhang[1]*

1 State Key Laboratory for Conservation and Utilization of Bio-Resources in Yunnan, College of Plant Protection, Yunnan Agricultural University, Kunming, China, 2 Yunnan Urban Agricultural Engineering and Technological Research Center, College of Agronomy and Life Sciences, Kunming University, Kunming, China, 3 Yunnan Expo Horticulture Company Limited, Kunming, China

* chenghkm@126.com (GC); zxmalex@126.com (XZ)

**Data Availability Statement:** All relevant data are within the paper and its Supporting Information files.

**Funding:** This work was supported by Yunnan Fundamental Research Projects [grant no.

## Abstract

Agricultural landscape pattern may enhance biocontrol services by supporting parasitoid populations, including parasitoids of *Bemisia tabaci* Gennadius (Hemiptera: Aleyrodidae). In this study, we selected four landscape types in Yunnan province, in south China, which were characterized by flower fields, mountain, river, and urban areas as their main elements. We then carried out systematic surveys in tomato fields found within each landscape type, to determine the diversity, occurrence, and parasitism rate of parasitoids. We found that parasitoids from the genus *Encarsia* and *Eretmocerus* were the main natural enemies present, and the most abundant species were recorded in the flower and the mountain landscapes. Also, *Encarsia formosa* (Hymenoptera: Chalcidoidea) formed the highest relative abundance regardless of the landscape type. We also found that the population density of *B. tabaci* in flower landscapes was the lowest than that in other landscape types. During the main activity period of *B. tabaci*, the parasitism rate in the flower landscape was higher than that in other landscape types. Our findings showed that the occurrence of *B. tabaci* was relatively mild in the flower landscapes. The landscape type was also beneficial to parasitoids as it enhanced their parasitism rate. Therefore, agriculture environmental schemes should consider increasing the size of flower fields in the surrounding landscape to enhance the sustainable control of *B. tabaci* by the natural agricultural ecosystem.

## Introduction

The whitefly, *Bemisia tabaci* Gennadius (Hemiptera: Aleyrodidae) is one of the most economically and agriculturally important insect pest worldwide [1]. It is a polyphagous species, with more than 500 plant species including tomato, cucumber, and other vegetables identified as it hosts [2, 3]. The pest cause serious economic damage not only by direct feeding but also by transmitting plant viruses such as *Begomovirus*, *Carlavirus*, *Crinivirus*, *Ipomovirus*, and *Torradovirus* [4]. The rapid expansion of whitefly populations promotes fast and efficient viral transmissions [5–7].

As an environmental-friendly control method, the use of parasitoids to control *B. tabaci* is not uncommon, in areas where consumers desire reduced use or complete elimination of pesticides

202201AT070269]; the National Natural Science Foundation of China [grant no. 31760541]; the Reserve Talent Project of Yunnan's Young and Middle-aged Academic and Technical Leaders [grant no. 202105AC160071]; the Young Top Talents of "High-level Talents Training Support Program in Yunnan Province" [grant no. YNWRQNBJ2020291]; and the Reserve Talents Project for the 17th Batch of Kunming's Young and Middle-aged Academic and Technical Leaders [grant no. KMRCH2019023]. One of the corresponding authors, Dr. Xiaoming Zhang, is the funder. He had role in Resources, Data Curation, Writing - Review and Editing, Funding acquisition.

**Competing interests:** The authors have declared that no competing interests exist.

from their food [8]. Aphelinid parasitoids, particularly *Encarsia formosa* and *Eretmocerus mundus* (Hymenoptera: Chalcidoidea), have outstanding records of successes in biological control against *B. tabaci* in many countries [9–11]. However, agricultural landscapes in China have changed from a complex pattern with a large proportion of natural habitats, to a simple landscape with a large proportion of arable land, which has dramatically changed the arrangement of arable and non-crop habitats [12, 13]. Agricultural landscape pattern can affect the interaction between pests and parasitoids [14]. First, the parasitism rate of parasitoids in agricultural landscape decreases with the increase in the proportion of crop farmland, because management does not aim to ensure overwintering and reproduction of the parasitoids [15]. Second, an increase in the proportion of non-crop habitats enhances the parasitism rate, because non-crop habitats provide wintering habitat and alternative hosts and food for parasitoids [16]. Third, in a diversified landscape, flowering plants in non-crop habitats can provide food sources such as pollen and nectar for parasitoids [17]. Structurally-complex landscapes can enhance biological control as a result of the high proportion of non-crop habitats, such as wooded mountain, grassland, hedgerows, and rivers in these landscapes [18]. These non-crop habitats may enhance species diversity or the abundance of insect parasitoids in nearby arable lands, thus improving the natural control of pests by effectively reducing their population [19, 20].

In our previous study, we found that although the parasitoids of *B. tabaci* could be observed in the tomato planting fields, the number of species and black pupae of the wasps were significantly difference under different agricultural landscapes around Kunming, Yunnan Province (S1 Data). Therefore, we hypothesized that the different agricultural landscapes may affect the species and control effect of parasitoids against whitefly. Because of the supply of food, shelter, alternative hosts, and favorable microclimates, parasitoids could benefit from natural field habitats around agricultural landscapes [21, 22]. Yunnan province in China is one of the regions with the richest biodiversity in the world [23, 24]. However, due to the intensification of human activities, the landscape in this province has changed significantly, which has tilted the original ecological balance and promoted the outbreak of the whitefly [25]. In this 2-year study, we selected four typical landscape types characterized with flowers, mountains, rivers, and urban areas as their main elements, in central Yunnan, where agricultural activities are most frequent. The focus of this study on the colonization of whitefly parasitoids in the different landscape types is considered critical for increasing the diversity of parasitoids to effectively suppress the pest population [20]. The study had two aims; the first was to assess the landscape types that best protect the abundance, occurrence, and enhance parasitism rate of parasitoids in these four typical landscape types. The second was to assess the variability of biocontrol services associated with the relationship between tomato growth stage, *B. tabaci* occurrence periods and parasitism efficiency (plot and landscape level) and also between time periods. Therefore, the overall aim was to identify which landscape type was most favorable to protect parasitoids and enhance their effective control to *B. tabaci*.

## Materials and methods

### Study area

The study was conducted within a radius of 0.5 km in agriculture landscapes around each of 12 tomato field plots (20 m×40 m), which were located in the surroundings of Kunming, south China (24˚42'45"N-25˚22'43"N, 102˚22'18"E-103˚10'90"E). The use of Google Earth Profession and field inspections (ground-truthing) were used to determine the land cover types [20, 26, 27]. A principal component analysis (PCA) was performed to reduce the dimensions of the landscape data. Ten land cover types were identified for the PCA analysis, the land cover type with the largest area and the absolute value of first principal component greater than 0.9 was selected as the landscape type: (1) flower fields, (2) river (rivers, lakes, reservoirs etc.), (3)

mountain (forest with altitude difference more than 150 m), (4) urban areas, (5) vegetable fields, (6) fruit trees, (7) trees (windbreaks, border trees or ornamental trees), (8) bushes, (9) grasslands, and (10) wastelands. The altitude difference among the landscape types was within 20 m, except for mountains. Principal component axes were extracted using correlations among the landscape types, and the resulting factors were not rotated.

1. Flower landscape type: three of them were divided into flower landscape type, their main landscape cover types were flower fields, which evenly distributed in the agricultural landscape types, and the main types of flowers were *Rosa chinensis*, *Dianthus caryophyllus*, *Myosotis sylvatica* and *Eustoma grandiflorum*.

2. River landscape type: three of them were divided into river landscape type, their main landscape cover types were rivers. The Panlong river across these three landscapes, the main type of tree plant was *Eucalyptus robusta*, and some fruits such as *Vitis vinifera* and *Malus domestica* and vegetables such as *Cucumis sativus* were planted here. The purpose of setting up the river landscape type was to pay attention to the high humidity environment caused by the river.

3. Mountain landscape type: three of them were divided into mountain landscape type, their main landscape cover types were cypress forest with altitude difference more than 150 m. There was abundant vegetation, the main types of tree plants were *Sabina chinensis* and *Pinus yunnanensis*, the main types of shrubby plants were *Cotoneaster microphyllus* and *Pyracantha fortuneana*, and the main types of herbaceous plants were *Imperata cylindrica*, *Polystichum acutidens* and *Cymbidium elegans*.

4. Urban landscape type: three of them were divided into urban landscape type. These landscapes were close to the town and their main landscape cover types were buildings. There were few vegetation species in this landscape type, and only some fruits such as *V. vinifera* and *M. domestica* and vegetables such as *C. sativus* and other vegetation are scattered.

Each plot was at least 5 km apart to avoid potential interactions of the insect populations from different sites. The study was carried out in tomato planting fields both in 2018 and 2019 under the different landscape types. The cultivar planted was tomato cv. 'Zhongyan TV1' (*Lycopersicon esculentum* Mill.). Tomato seedlings were first nursed in June and harvested in October. No plot was treated against pests or diseases in our experiments. Plots were kept weed-free by manual weeding when necessary.

## Sampling

**Population dynamics.** The population dynamics of *B. tabaci* and its parasitoids was determined by sampling in each 800m² plot (20 m×40 m) of tomato planting field in each landscape. The first sampling started 10 days after tomato transplanting. Five yellow sticky traps (20 cm×25 cm) at each sampling point were hanged in the fields and collected every 10 days using the same five point sampling method until the end of the growing season, the five yellow sticky traps were positioned following the planting line of tomato, and the height increased with the growth of tomato. The yellow sticky traps were taken back to the laboratory to count the number of individuals of *B. tabaci* and its parasitoids. The averages of the total numbers of *B. tabaci* and its parasitoids on each yellow sticky trap were calculated as their respective population densities. The adult parasitoids were collected for species identification. The growth period of tomato was recorded at each survey. This was divided into seedling, anthesis, fruit expansion, and harvest periods [28, 29].

**Relative abundance of parasitoids.** The number of individual species in the community obtained from each yellow sticky trap was used as the basis for data analysis, the relative abundance of each parasitoid species was calculated as the proportion [30].

**Parasitism rate.** Parasitism rates were evaluated in the same tomato planting field alongside the population dynamics. The first sampling started 10 days after tomato transplanting. Leaves with nymphs of *B. tabaci* were collected every 10 days using the five point sampling method until the end of the growing season. In each sampling point, five tomato plants were sampled (avoiding the plants closest to any edge to minimise edge effects). On each tomato plant, five leaves of similar age at the upper, middle, and lower positions were examined, giving a total of 375 monitored leaves per field. Leaves were transported to the laboratory to count nymphs, after which they were placed in a Petri dish with agar and kept in an artificial climate box (Shanghai Boxun, BIC-400, T = 25˚C, rH = 65%, L/D = 14h/10h). The parasitism rate was calculated after eclosion of *B. tabaci* and parasitoids based on the formula [31, 32]:

$$P = P_e \div (W_e + P_e) \times 100,$$

where $P$ is the percentage of parasitism, $P_e$ is the number of parasitoids, and $W_e$ is the number of *B. tabaci*.

### Description of seasonal activity

The seasonal activity curve was standardized following the quartile method of Fazekas et al. and Zhang et al. [25, 33]. This method divides the seasonal activity into three periods: early, main, and late, and formally identifies the start and end of each of these, as well as the date of the seasonal activity peak. First, the numbers observed are summed and the three cardinal points are the dates when 25, 50 and 75% of the total densities are reached. These also divide the curve into four segments. The start of the main activity period corresponds to the date when the cumulative densities reach 25% of the total (the start of the second quartile on the vertical axis), and the end corresponds to the date when 75% (the end of the third quartile on the vertical axis) is reached. The early activity period was defined formally as from the start of the census to the beginning of the main activity period, and the likewise formalized late activity period was defined formally as from the end of the main activity period until the end of the observations, when activity stopped.

### Data analysis

The population densities of *B. tabaci* and *E. formosa* adults (the average quantity of adults in each yellow sticky trap), as well as the relative abundance of parasitoids and the parasitism rate of *E. formosa* on *B. tabaci*, were analyzed by using a one-way analysis of variance (ANOVA) after tests of normality (Shapiro–Wilk) and homoscedasticity (Bartlett), with agriculture landscape types as the main effect. The data have been logarithmic transformed if it did not follow a normal distribution. To reduce the impact of occurrence time on the population densities of *B. tabaci* and its parasitoids, the activity period of *B. tabaci* and its parasitoids were divided into early, main, and late activity period by quartile method. The least significant differences (LSD) were determined using data from one sampling conducted in the same activity periods, in the same agriculture landscape types as one replicate. A significance level of $P = 0.05$ was used for all tests. Data analyses were performed using SPSS 20.0. The figures of cumulative seasonal activity curves and population dynamics were made with Origin 2018.

## Results

### Interpretation of principal components

At each landscape, we were able to interpret the first three principal components which accounted for about 85% of the variation in the landscape variables. We divided the 12 landscapes into four types according to the interpretation of principal components. Three of these which had the

highest eigenvector of principal component 1 of flower fields (2018: 0.963, 0.950, and 0.913, respectively; 2019: 0.940, 0.985, and 0.944, respectively), were grouped as flower landscape type. Another three which had the highest eigenvector of principal component 1 of river (2018: -0.956, -0.992, and 0.938, respectively; 2019: -0.970, -0.992, and -0.921, respectively), were grouped as river landscape type. The next three which had the highest eigenvector of principal component 1 of mountain (2018: -0.945, 0.966, and 0.915, respectively; 2019: 0.915, 0.900, and -0.920, respectively) were grouped as mountain landscape type. The last three which had the highest eigenvector of principal component 1 of urban (2018: 0.946, -0.906, and 0.927, respectively; 2019: -0.907, 0.911, and -0.913, respectively) were grouped as urban landscape type (Table 1).

## Parasitoid species of *Bemisia tabaci* in different types of landscapes

More than 150,000 parasitoids were collected in the sampled landscapes; they belonged to the genus *Encarsia* and *Eretmocerus*. The number of parasitoid species in the flower and the mountain landscapes were higher than those in the river and the urban landscapes. *E. formosa* accounted for the highest relative abundance in the four landscape types regardless of the tomato planting years (flower landscape in 2018: $F = 43.0190$; $df = 5, 17$; $P = 0.0001$. river landscape in 2018: $F = 266.1050$; $df = 5, 17$; $P = 0.0001$. mountain landscape in 2018: $F = 63.4520$; $df = 5, 17$; $P = 0.0001$. urban landscape in 2018: $F = 151.6260$; $df = 5, 17$; $P = 0.0001$. flower landscape in 2019: $F = 144.3720$; $df = 5, 17$; $P = 0.0001$. river landscape in 2019: $F = 901.3930$; $df = 5, 17$; $P = 0.0001$. mountain landscape in 2019: $F = 303.2620$; $df = 5, 17$; $P = 0.0001$. urban

**Table 1. Principal component loading diagrams examining the landscape variables at four landscape types.**

| Planting years | Landscapes | Eigenvector of Principal Component 1 | | | | | | | | | |
|---|---|---|---|---|---|---|---|---|---|---|---|
| | | flower fields | river | mountains | urbans | vegetable fields | fruit trees | trees | bushes | grasslands | wastelands |
| 2018 | 1 | 0.96 | 0.23 | 0.04 | 0.78 | -0.67 | 0.32 | -0.40 | -0.41 | 0.80 | -0.86 |
| | 2 | 0.95 | -0.31 | -0.55 | 0.87 | 0.58 | 0.04 | 0.61 | -0.38 | -0.06 | -0.85 |
| | 3 | 0.91 | -0.33 | 0.34 | 0.07 | 0.24 | -0.80 | 0.89 | 0.57 | 0.74 | -0.84 |
| | 4 | 0.77 | -0.96 | 0.42 | 0.36 | -0.82 | 0.71 | 0.29 | 0.48 | -0.83 | 0.20 |
| | 5 | 0.78 | -0.99 | -0.44 | -0.46 | -0.62 | 0.80 | 0.74 | 0.38 | -0.48 | 0.89 |
| | 6 | -0.89 | 0.94 | 0.66 | 0.73 | -0.86 | 0.39 | 0.29 | -0.05 | 0.53 | -0.79 |
| | 7 | -0.43 | -0.43 | -0.95 | 0.78 | 0.86 | 0.88 | -0.06 | 0.27 | 0.29 | -0.51 |
| | 8 | 0.64 | 0.82 | 0.97 | -0.31 | 0.49 | 0.18 | -0.46 | 0.87 | -0.10 | -0.49 |
| | 9 | 0.43 | -0.46 | 0.92 | 0.16 | 0.80 | 0.75 | 0.22 | -0.55 | -0.22 | -0.88 |
| | 10 | 0.74 | 0.89 | -0.52 | 0.95 | -0.52 | -0.68 | 0.83 | 0.63 | -0.34 | -0.87 |
| | 11 | 0.24 | -0.60 | 0.76 | -0.91 | 0.72 | -0.80 | 0.83 | -0.87 | -0.22 | 0.89 |
| | 12 | 0.81 | -0.41 | 0.77 | 0.93 | -0.33 | 0.20 | 0.61 | -0.71 | 0.19 | -0.89 |
| 2019 | 1 | 0.94 | 0.16 | 0.64 | 0.60 | -0.14 | 0.87 | 0.87 | 0.74 | -0.73 | -0.84 |
| | 2 | 0.99 | 0.40 | 0.25 | 0.49 | -0.40 | -0.22 | 0.68 | 0.76 | -0.74 | -0.79 |
| | 3 | 0.94 | 0.71 | -0.53 | 0.14 | -0.53 | -0.53 | 0.41 | 0.70 | -0.77 | -0.53 |
| | 4 | 0.87 | -0.97 | 0.82 | -0.70 | -0.48 | 0.73 | 0.62 | 0.76 | -0.75 | 0.88 |
| | 5 | 0.89 | -0.99 | -0.63 | -0.54 | -0.50 | 0.80 | 0.68 | 0.84 | -0.62 | 0.86 |
| | 6 | 0.87 | -0.92 | 0.71 | -0.67 | -0.75 | 0.81 | 0.73 | 0.72 | -0.70 | 0.86 |
| | 7 | -0.39 | 0.32 | 0.92 | 0.86 | 0.69 | 0.50 | 0.22 | 0.33 | -0.72 | -0.86 |
| | 8 | -0.45 | -0.84 | 0.90 | 0.70 | 0.49 | 0.46 | 0.66 | 0.35 | -0.86 | -0.71 |
| | 9 | -0.49 | -0.32 | -0.92 | 0.81 | 0.47 | 0.21 | 0.42 | -0.16 | -0.75 | 0.88 |
| | 10 | -0.39 | -0.54 | 0.65 | -0.91 | 0.65 | -0.87 | 0.83 | -0.76 | 0.88 | 0.89 |
| | 11 | 0.26 | 0.86 | -0.68 | 0.91 | -0.66 | 0.86 | -0.83 | 0.74 | -0.32 | -0.89 |
| | 12 | 0.85 | -0.62 | 0.72 | -0.91 | 0.73 | -0.83 | 0.81 | -0.87 | -0.25 | 0.89 |

**Table 2. Parasitoid species of *Bemisia tabaci* in different types of landscapes in Kunming, south China.**

| Planting years | Parasitoid species | Relative abundance in different landscape types (%) | | | |
|---|---|---|---|---|---|
| | | Flower landscape | River landscape | Mountain landscape | Urban landscape |
| 2018 | *Encarsia formosa* | 54.58±3.11a | 75.40±3.14a | 63.57±6.05a | 76.84±4.25a |
| | *Encarsia sophia* | 12.80±2.36b | 13.49±1.51b | 10.97±1.99b | 13.97±3.46b |
| | *Encarsia* sp1 | 11.71±5.45b | 4.40±2.56cd | 8.97±2.45b | 8.40±2.33b |
| | *Eretmocerus hayati* | 4.60±0.69b | 5.63±0.44c | 4.69±0.96b | 0.00±0.00c |
| | *Eretmocerus* sp1 | 7.04±0.70b | 0.00±0.00d | 7.00±1.62b | 0.00±0.00c |
| | Other species | 9.27±1.86b | 1.09±0.56cd | 4.80±0.63b | 0.79±0.48c |
| 2019 | *Encarsia formosa* | 57.58±2.41a | 79.37±1.41a | 71.92±1.15a | 84.30±2.08a |
| | *Encarsia sophia* | 12.94±1.67b | 12.87±1.22b | 10.37±2.67b | 10.68±1.36b |
| | *Encarsia* sp1 | 11.61±2.49b | 4.09±1.46c | 5.53±1.19c | 3.76±0.88c |
| | *Eretmocerus hayati* | 8.27±0.78bc | 1.41±0.70cd | 2.89±0.55c | 0.00±0.00d |
| | *Eretmocerus* sp1 | 3.32±0.78c | 0.00±0.00d | 4.14±1.41c | 0.00±0.00d |
| | Other species | 6.29±1.10c | 2.26±0.55cd | 5.15±1.57c | 1.25±0.59cd |

Data in the table are mean ± SE. The different lowercases indicate significantly different at the 0.05 level with the different parasitoid species in the same type of landscapes during the same tomato's planting year.

landscape in 2019: *F* = 913.9680; *df* = 5, 17; *P* = 0.0001), indicating that *E. formosa* was the main parasitoid in the four landscape types (Table 2).

## Seasonal activity of *Bemisia tabaci* and *Encarsia formosa*

The length of the main activity periods of *B. tabaci* in 2018 ranged from 21 days (flower landscape and river landscape) to 23 days (mountain landscape and urban landscape) and the peak activity was reached on 3rd September (flower landscape), 28th August (river landscape), 18th September (mountain landscape) and 19th October (urban landscape). The length of the main activity periods of *B. tabaci* was not significantly different between years, and it ranged from 21 days (urban landscape) to 31 days (mountain landscape) in 2019. The peak activity was reached on early September to mid-September. For *E. formosa*, the length of the main activity periods in 2018 was 47 days in the urban landscape and about 30 days in other landscape types. The peak activity was reached on 9th September (flower landscape), 12th September (river landscape), 23rd September (mountain landscape) and 19th October (urban landscape). The length of the main activity periods in 2019 was 22 days in the flower landscape and about 30 days in the other landscape types. The peak activity was reached on 10th September (flower landscape and river landscape) and 20th September (mountain landscape and urban landscape). The main activity period of *B. tabaci* and *E. formosa* encompassed the anthesis to fruit expansion of tomato in both years of study (Table 3).

## Population dynamics of *Bemisia tabaci* and *Encarsia formosa*

In the flower landscape, the population densities of *B. tabaci* and *E. formosa* were relatively low in the first survey in 2018, then they all gradually increased. The population density of *B. tabaci* fluctuated after late July, increased at a sharp rate in late August and peaked in early September (female: 70.80 per. yellow sticky trap; male: 32.40 per. yellow sticky trap). The population density of *E. formosa* increased from the beginning of the survey to late August, then increased to the maximum population density in late September after a slight decline (39.40 per. yellow sticky trap). During the survey of 2019, the population of *B. tabaci* maintained a relatively low trend till the mid-August,

**Table 3. Main activity periods and peak activity dates of *Bemisia tabaci* adults and *Encarsia formosa* in different types of landscapes in Kunming, south China.**

| Planting years | Landscape types | *Bemisia tabaci* | | *Encarsia formosa* | |
|---|---|---|---|---|---|
| | | **Main activity period (duration in days)** | **Peak activity date** | **Main activity period (duration in days)** | **Peak activity date** |
| 2018 | Flower landscape | 21 Aug.-10 Sep. (21) | 03 Sep. | 21 Aug.-19 Sep. (30) | 09 Sep. |
| | River landscape | 21 Aug.-10 Sep. (21) | 28 Aug. | 30 Aug.-28 Sep. (30) | 12 Sep. |
| | Mountain landscape | 07–29 Sep. (23) | 18 Sep. | 07 Sep.-09 Oct. (31) | 23 Sep. |
| | Urban landscape | 08–30 Oct. (23) | 19 Oct. | 27 Sep.-13 Nov. (47) | 19 Oct. |
| 2019 | Flower landscape | 20 Aug.-10 Sep. (22) | 03 Sep. | 30 Aug.-20 Sep. (22) | 10 Sep. |
| | River landscape | 20 Aug.-10 Sep. (22) | 01 Sep. | 30 Aug.-30 Sep. (32) | 10 Sep. |
| | Mountain landscape | 30 Aug.-20 Sep. (31) | 10 Sep. | 30 Aug.-30 Sep. (32) | 20 Sep. |
| | Urban landscape | 10–30 Sep. (21) | 16 Sep. | 30 Aug.-30 Sep. (32) | 20 Sep. |

and then the density of female adults began to increase gradually and peaked to 68.20 per. yellow sticky trap in early September. The density of male adults maintained a relatively flat trend till the end of the sample date. During the same sampling period, the population of *E. formosa* also maintained a relatively low trend till the middle august, then increased gradually and peaked to 35.73 per. yellow sticky trap in late September and then decreased till the end of the sample date (Fig 1).

In the river landscape, the population density of *B. tabaci* increased from early July to mid-July and then increased sharply from late July after a brief decrease. The density of female adults peaked to 334.13 per. yellow sticky trap in mid-September, and the density of male adults peaked to 231.20 per. yellow sticky trap in late August. After that, the population declined sharply till the end of the investigation in 2018. From the beginning of the survey, the population of *E. formosa* kept a slow growth trend till mid-early September, then decreased a little till late September. It began to rise again, peaked to 36.07 per. yellow sticky trap in early October, and then declined till the end of the sample date in 2018. During the survey in 2019, the population density of *B. tabaci* began to increase gradually from early July to late July, then they increased sharply and peaked in early September (female: 301.60 per. yellow sticky trap; male: 182.13 per. yellow sticky trap). After that, the population density of *B. tabaci* decreased sharply till the end of the sample date. The population of *E. formosa* began to increase gradually from early July and peaked to 38.27 per. yellow sticky trap in mid-August, then they slowly declined to mid-September and rise again to 34.33 per. yellow sticky trap in mid-October and then slowly declined till the end of the sample date (Fig 1).

In the mountain landscape, the population density of *B. tabaci* increased slowly from mid-July till mid-late August, reached its first peak in early September (female: 118.53 per. yellow sticky trap; male: 60.13 per. yellow sticky trap), then declined briefly. It increased to its second peak in late September (female: 166.13 per. yellow sticky trap; male: 71.47 per. yellow sticky trap), then it declined sharply till the end of the sample date in 2018. The population of *E. formosa* increased steadily from the beginning of the survey, and then slowly increased to the maximum population density (36.53 per. yellow sticky trap) in mid-late October after a small decline in early September. After that, it declined till the end of the sample date in 2018. During the survey in 2019, the population density of *B. tabaci* began to increase gradually from early July. The density of female adults increased sharply from early August and peaked to 190.40 per. yellow sticky trap in late September. The density of male adults increased sharply from late August to early September, then increased slowly and peaked to 69.67 per. yellow sticky trap in late September. After that, the population density of *B. tabaci* decreased till the end of the sample date. The population of *E. formosa* maintained a relatively low trend till the early October, then increased and peaked to 40.67 per. yellow sticky trap in mid-October. After that, it slowly declined till the end of the sample date (Fig 1).

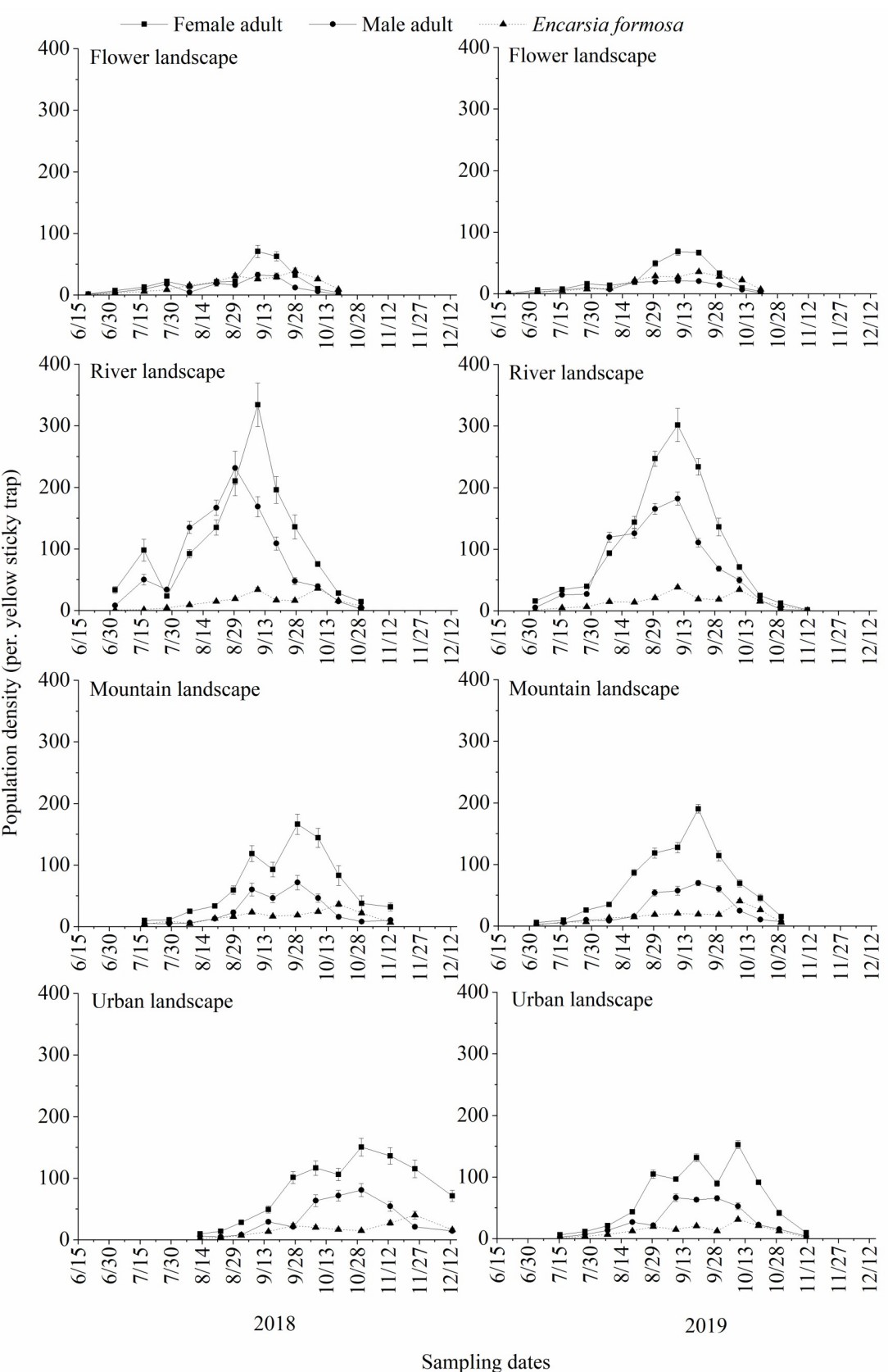

**Fig 1. Population dynamics of *Bemisia tabaci* adults and *Encarsia formosa* (mean±SE) in different types of landscapes in Kunming, south China.**

In the urban landscape, the population density of *B. tabaci* and *E. formosa* were relatively low in July. In early August, the population of *B. tabaci* began to increase, and both female and male adult densities peaked in late October (female: 150.40 per. yellow sticky trap; male: 80.87 per. yellow sticky trap). The population then declined slowly till the end of the sample date in 2018. The population of *E. formosa* gradually increased from mid-late August to late September, decreased to late October and then began to rise again, peaking to 40.27 per. yellow sticky trap in mid-late November. The population then declined till the end of the sample date in 2018. During the survey in 2019, the population density of *B. tabaci* maintain a relatively flat and low trend till to mid-August. The density of female adults began to increase sharply from mid-August to late August, then fluctuated and peaked to 152.67 per. yellow sticky trap in mid-October. The density of male adults began to increase sharply from late August and peaked to 66.47 per. yellow sticky trap in mid-September, then maintain a relatively flat trend till to mid-October. After that, the population density decreased till the end of the sample date. The population of *E. formosa* gradually increased to late August and maintain a relatively flat trend till to late September, then increased and peaked to 31.00 per. yellow sticky trap in mid-October. The population then declined till the end of the sample date (Fig 1).

There were variations between the population dynamics of *B. tabaci* and *E. formosa* in different types of landscapes. That of *B. tabaci* in flower landscapes was lower than in other landscape types, followed by the urban and the mountain landscapes in which the population dynamics changed significantly. The population dynamics in river landscape was more evident. The growth trend was faster, with a clear peak. The population dynamics trend of *E. formosa* was generally the same in different landscape types (Fig 1).

The highest population density of *B. tabaci* during the early and main activity periods, was recorded in the river landscape compared to the other landscape types. The population density in the flower landscape was significantly lower than that in the other landscapes during the main activity period (early activity period in 2018: $F = 9.9230$; $df = 3, 56$; $P = 0.0001$. early activity period in 2019: $F = 6.3000$; $df = 3, 56$; $P = 0.0010$. main activity period in 2018: $F = 43.1660$; $df = 3, 35$; $P = 0.0001$. main activity period in 2019: $F = 42.8840$; $df = 3, 35$; $P = 0.0001$). During the late activity period, the population density in the flower landscape was significantly lower than that in the urban and the river landscapes in 2018 ($F = 3.5920$; $df = 3, 47$; $P = 0.0209$), and was significantly lower than that in the river landscapes in 2019 ($F = 2.1200$; $df = 3, 53$; $P = 0.1094$). During the early and late activity periods of *E. formosa*, there was no significant difference in the population density among the four types of landscapes (early activity period in 2018: $F = 1.2630$; $df = 3, 56$; $P = 0.2964$. late activity period in 2018: $F = 0.5460$; $df = 3, 32$; $P = 0.6550$. early activity period in 2019: $F = 0.4260$; $df = 3, 59$; $P = 0.7351$. late activity period in 2019: $F = 1.0900$; $df = 3, 41$; $P = 0.3649$). During the main activity period, the density of *E. formosa* in the flower landscape was significantly higher than that in the other three types of landscapes (2018: $F = 3.1700$; $df = 3, 50$; $P = 0.0328$. 2019: $F = 9.9140$; $df = 3, 44$; $P = 0.0001$) (Table 4).

## Parasitism rates in different types of landscapes

There was no significant difference in parasitism rate of *B. tabaci* in different types of landscapes during the early activity period in 2018 ($F = 0.3720$; $df = 3, 62$; $P = 0.7732$). Parasitism rate was higher in flower landscape than that in urban landscape in 2019 ($F = 2.1650$; $df = 3, 56$; $P = 0.1030$). During the main activity period, the parasitism rate was highest in flower landscape and lowest in urban landscape in 2018 ($F = 46.8010$; $df = 3, 35$; $P = 0.0001$). The

**Table 4. Population density comparison of *Bemisia tabaci* and *Encarsia formosa* in different types of landscapes in Kunming, south China.**

| Planting years | Species | Landscape types | Population density (per. yellow sticky trap) | | |
|---|---|---|---|---|---|
| | | | Early activity period | Main activity period | Late activity period |
| 2018 | *Bemisia tabaci* | Flower landscape | 18.84±3.43c | 60.47±11.68c | 39.30±10.47b |
| | | River landscape | 118.45±26.10a | 415.16±39.44a | 132.41±30.94a |
| | | Mountain landscape | 38.29±7.00bc | 185.16±15.35b | 94.43±20.92ab |
| | | Urban landscape | 54.17±11.05b | 196.31±9.27b | 137.40±18.42a |
| | *Encarsia formosa* | Flower landscape | 6.64±1.51a | 26.80±1.42a | 24.73±4.48a |
| | | River landscape | 6.07±1.46a | 21.60±2.46b | 19.82±4.39a |
| | | Mountain landscape | 9.52±1.37a | 20.63±1.08b | 21.73±4.33a |
| | | Urban landscape | 6.45±1.36a | 20.36±1.47b | 28.17±5.69a |
| 2019 | *Bemisia tabaci* | Flower landscape | 13.84±2.47c | 65.40±7.91c | 39.28±9.70b |
| | | River landscape | 90.33±22.21a | 388.53±36.50a | 121.40±29.92a |
| | | Mountain landscape | 41.15±9.07bc | 205.89±15.13b | 86.78±17.17ab |
| | | Urban landscape | 51.48±11.57b | 170.96±8.04b | 97.47±21.87ab |
| | *Encarsia formosa* | Flower landscape | 7.77±1.74a | 30.47±1.65a | 19.27±3.28a |
| | | River landscape | 8.28±1.44a | 24.30±2.79b | 14.58±3.84a |
| | | Mountain landscape | 8.53±1.37a | 19.23±0.67bc | 24.24±5.22a |
| | | Urban landscape | 6.10±1.39a | 16.77±1.44c | 16.53±3.34a |

Data in the table are mean ± SE. The different lowercases indicate significantly different at the 0.05 level in different types of landscapes with the same insect during the same tomato's planting year.

parasitism rate was highest in flower landscape and lowest in urban landscape in 2019 ($F = 9.1890$; $df = 3, 35$; $P = 0.0002$). During the late activity period, the parasitism rates in mountain landscape was higher than that in river and in urban landscapes in 2018 ($F = 2.8360$; $df = 3, 44$; $P = 0.0498$). There was no significant difference among the four landscape types in 2019 ($F = 1.2200$; $df = 3, 53$; $P = 0.3123$) (Table 5).

## Discussion

### Parasitoid species of *Bemisia tabaci* in different types of landscapes

Our results showed that the main natural enemies of *Bemisia tabaci* belonged to *Encarsia* and *Eretmocerus* except for those in the urban landscape. Qiu *et al*. and Li *et al*. also reported that

**Table 5. Parasitism rates in different types of landscapes in Kunming, south China.**

| Planting years | Landscape types | Parasitism rate (%) | | |
|---|---|---|---|---|
| | | Early activity period | Main activity period | Late activity period |
| 2018 | Flower landscape | 19.78±4.96a | 58.43±3.17a | 21.68±6.20a |
| | River landscape | 18.78±3.65a | 21.38±1.87c | 9.80±1.31b |
| | Mountain landscape | 18.45±3.15a | 37.50±4.23b | 23.79±2.70a |
| | Urban landscape | 15.18±2.51a | 13.88±1.34c | 14.57±4.25ab |
| 2019 | Flower landscape | 20.94±2.23a | 52.66±7.11a | 22.60±3.73a |
| | River landscape | 16.34±3.37ab | 23.34±1.91bc | 17.46±3.65a |
| | Mountain landscape | 16.03±2.25ab | 33.26±4.62b | 19.94±3.25a |
| | Urban landscape | 11.94±2.55b | 18.74±4.80c | 12.87±2.98a |

Data in the table are mean ± SE. The different lowercases indicate significantly different at the 0.05 level in different types of landscapes during the same activity period and the same tomato's planting year.

the parasitoids of *B. tabaci* mainly belonged to *Encarsia* and *Eretmocerus* [31, 34]. In our study, the abundance of parasitoids was high in the flower and the mountain landscapes. Many studies have reported that flowering plants and abundant vegetation are beneficial to the diversity of natural enemies. Flowers in the flower landscape provided enough additional food sources for parasitoids, such as nectar. The rich vegetation in the mountain landscape also likely provided good shelter from agricultural disturbance and overwintering sites [35–37].

## Population dynamics of *Bemisia tabaci* and *Encarsia formosa*

In the flower landscape, the population of *B. tabaci* was always significantly lower than that in the other landscape types. The presence of non-host plant plants species of *B. tabaci* such as *Rosa chinensis*, *Dianthus caryophyllus*, *Myosotis sylvatica* and *Eustoma grandiflorum* in the landscape could be the influencing factor [34]. On the other hand, the presence of abundant flowering plants in the landscape, may have improved the control ability of the parasitic natural enemies [35, 38, 39]. The combined action of the lots of non-host plant species and parasitic natural enemies kept the population of *B. tabaci* at a low density. Xiao *et al.* planted papaya (*Carica papaya* L.) and other banker plants beside a field to enhance the efficacy of biocontrol, and this had an enhancing effect on the population of *Encarsia sophia* [11]. Therefore, the use of flowering plants as banker plants beside tomato fields, can provide energy substrates for parasitoids to increase their parasitism rate and population densities, for effective biological control.

Our results showed that the growth trend of *B. tabaci* in tomato plantings in the river landscape was faster than that in other landscape types, and the population density was higher. The existence of the river created a higher humidity over the landscape. The flight activities of some pests under high humidity conditions are significantly higher than that under low humidity conditions, which may be related to reduced water evaporation under low humidity conditions [40–42]. Similarly, humidity also significantly affects the growth, development, survival, and longevity of pests [43, 44]. Therefore, high humidity may be one of the main reasons for the outbreak of *B. tabaci* in the landscape.

## Parasitism rate in different types of landscapes

During the main activities period, the parasitism rate in tomato plantings in the flower landscape was higher, while that in the urban and the river landscapes were lower. There were abundant flowering plants in the flower landscape, which may have provided abundant nectar. Most parasitic natural enemies need to supplement their nutrition by feeding on pollen and nectar to promote the development of their reproductive system, especially the ovary [38, 45, 46], thereby increasing the number of eggs laid [35, 47] and parasitism rate [39, 48]. Similarly, carbohydrates in pollen and nectar can also provide energy for the survival and activity of natural enemies and prolong their life span [49]. Flower landscapes provide these energy substances. Therefore, it is necessary to set up flowering plants in farmland landscape. However, different flowering plants have different structures and signals towards natural enemies, which results in different levels of attraction. Similarly, different natural enemies show selectivity to different flowering plants due to their different nutritional needs [50–52]. For instance, the presence of different plants around a field had different effects on the parasitism rate of whitefly parasitoids [53]. In our study, we recorded many kinds of flowering plants in the flower landscape. It is therefore necessary to conduct a further study, to identify which of these plants is most beneficial to the fitness of the parasitoids.

In summary, this study aimed to study the impact of farmland landscape on parasitic natural enemies and pests. Together, these can be considered as an agricultural ecosystem. In

addition to parasitic natural enemies, predatory natural enemies and entomopathogenic fungi also control *B. tabaci* [34]. Therefore, this requires that consideration is given to them in the analysis of the natural enemy complex. Furthermore, the role of neutral insects in farmland ecosystems should also be considered, as they serve as alternative food for natural enemies and play important roles in maintaining ecosystem stability [54]. The need to establish a sustainable green eco-agricultural landscape requires that studies are continually carried out to assess the feasibility of green development of different types of landscape.

## Supporting information

**S1 Data. Previous study, parasitoid species of Bemisia tabaci and its population densities.** Data were initially subjected to a one-way ANOVA with agriculture landscape variables as the main effect. Difference in population densities was compared among agriculture landscape types by Least Significant Difference. The significance threshold was $P = 0.05$ in all tests. Data analyses were performed using SPSS 20.0.
(XLSX)

**S2 Data. Metadata, covariance matrix of PCA.** The 12 agriculture landscapes located in the surroundings of Kunming, south China (24˚42'45"N-25˚22'43"N, 102˚22'18"E-103˚10'90"E). it was selected by use of Google Earth Profession and field inspections (ground-truthing) once a month during the tomato growing seasons in 2018 and 2019. The cover types in each landscape were divided into 10 types according to vegetation type, human factor interference and land type characteristics. A Principal Components Analysis (PCA) was performed to reduce the dimensions of the landscape data. These ten land cover types were divided for the PCA analysis, the land cover type with the largest area in one landscape and the absolute value of first principal component greater than 0.9 was selected as the landscape type. Principal component axes were extracted using correlations among variables, and the resulting factors were not rotated.
(XLSX)

**S3 Data. Parasitoid species of Bemisia tabaci and its relative abundance in different types of landscapes in Kunming, south China.** Census data were initially subjected to a one-way ANOVA with agriculture landscape types as the main effect. Differences in relative abundance were compared among different parasitoid species of *B. tabaci* in the same agriculture landscape types by Least Significant Difference. The significance threshold was $P = 0.05$ in all tests. Data analyses were performed using SPSS 20.0.
(XLSX)

**S4 Data. Population densities of Encarsia formosa and Bemisia tabaci.** Census data were initially subjected to a one-way ANOVA with agriculture landscape types as the main effect. To reduce the impact of occurrence time on the population densities of *E. formosa* and *B. tabaci*, the activity period of *E. formosa* and *B. tabaci* densities were divided into early, main and late activity period by quartile method. Differences in *E. formosa* and *B. tabaci* densities and parasitism rate were compared among agriculture landscape types in the same activity period by Least Significant Difference. The significance threshold was $P = 0.05$ in all tests. Data analyses were performed using SPSS 20.0.
(XLSX)

## Acknowledgments

We thank Dr. Yi-Bo Zhang (Chinese Academy of Agricultural Sciences) for help us at identification of parasitoid species and XiaoQingChong English Editing Services for editing the

English of the manuscript. We thank Wen-Jing Duan, Xiao-Yun Wang, Jian-Wen Lv, and Fei Wang (Yunnan Agricultural University of China) for help us at sampling work. We also thank Mr. Jun-Yao Liu, Yong-Hua Yang, Yin-Long Wu, and Ms. Feng-Ping Duan for help us at field management.

## Author Contributions

**Data curation:** Wenjun Dou, Mingjiang Li, Ziliao Wang, Guohua Chen, Xiaoming Zhang.

**Formal analysis:** Shaowu Yang.

**Funding acquisition:** Xiaoming Zhang.

**Investigation:** Shaowu Yang, Wenjun Dou, Mingjiang Li, Ziliao Wang.

**Methodology:** Shaowu Yang.

**Project administration:** Guohua Chen.

**Resources:** Guohua Chen, Xiaoming Zhang.

**Supervision:** Guohua Chen.

**Writing – original draft:** Shaowu Yang.

**Writing – review & editing:** Guohua Chen, Xiaoming Zhang.

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
