## [Decision Letter · Decision Letter 0]

28 Mar 2022

PONE-D-21-35756Flowering Agricultural Landscapes Enhance Parasitoid Biological Control to Bemisia tabaci on Tomato in South ChinaPLOS ONE

Dear Dr. Yang,

Thank you for submitting your manuscript to PLOS ONE. After careful consideration, we feel that it has merit but does not fully meet PLOS ONE’s publication criteria as it currently stands. Therefore, we invite you to submit a revised version of the manuscript that addresses the points raised during the review process.

 Please improve the manuscript according to the comments of the two reviewers.

We look forward to receiving your revised manuscript.

Kind regards,

Yonggen Lou

Academic Editor

PLOS ONE

Journal Requirements:

This work was supported by the National Natural Science Foundation of China [31760541]; the Reserve Talent Project of Yunnan’s Young and Middle-aged Academic and Technical Leaders [202105AC160071]; the Young Top Talents of “High-level Talents Training Support Program in Yunnan Province” [YNWRQNBJ2020291]; and the Reserve Talents Project for the 17th Batch of Kunming’s Young and Middle-aged Academic and Technical Leaders [KZF〔2019〕No. 43].

4. We note that you have referenced (ie. Bewick et al. [5]) which has currently not yet been accepted for publication. Please remove this from your References and amend this to state in the body of your manuscript: (ie “Bewick et al. [Unpublished]”) as detailed online in our guide for authors

Reviewers' comments:

Reviewer's Responses to Questions

**Comments to the Author**

1. Is the manuscript technically sound, and do the data support the conclusions?

Reviewer #1: Yes

Reviewer #2: Yes

2. Has the statistical analysis been performed appropriately and rigorously? 

Reviewer #1: Yes

Reviewer #2: Yes

3. Have the authors made all data underlying the findings in their manuscript fully available?

Reviewer #1: Yes

Reviewer #2: No

4. Is the manuscript presented in an intelligible fashion and written in standard English?

Reviewer #1: Yes

Reviewer #2: Yes

5. Review Comments to the Author

Reviewer #1: This manuscript investigates the effect of different landscape on occurrence and dynamics of whitefly Bemisia tabaci and its main parasitic wasps. The authors found that landscape dominated by flower field had lower whitefly population and higher natural enemies. Moreover, the parasitism rate of Encarsia formosa, the main species of whitefly natural enemy is higher in flower field landscape during the main activity period. The results indicate the floral resource may enhance the control potential of whitefly parasitoids and the landscape manipulation should be considered for sustainable control of the pest whitefly.

1. The word abondance should be abundance

2. L92: add “each of” before 12 tomato field plots

3. L320-321: This sentence is not clear to me

4. The biodiversity is one key factor influencing the landscape. The authors may have a brief description of the diversity on each of the landscape type. For example, what is the main flower in the flower field landscape? How about other landscape?

5. The predatory natural enemy is always co-occurrence with parasitic natural enemy. How about the role of predatory natural enemies in the field trial sites?

Reviewer #2: This manuscript describes the effects of flowering agricultural landscapes on Parasitoid Biological Control to Bemisia tabaci on Tomato in South China. The results are helpful for enhancing the sustainable control of B. tabaci in natural agro-ecosystems. However, there are still some problems in this manuscript and minor revisions are needed.

1. The hypothesis is based on finding that the parasitoids of B. tabaci could be observed in the tomato planting fields, but the number of species and black pupae of the wasps were significantly difference under different agricultural landscapes around Kunming, Yunnan Province. However, this is not published data, and it is better to put the content in the manuscript as supplementary material.

2. Some of the descriptions in the data analysis are confusing. Are the data normally distributed? The data should be transformed if it did not follow a normal distribution. Why use the LSD method instead of Tukey's HSD?

3. Some parts of the results lack the specific description of the statistical analysis results. For example, the comparison of relative abundance and population dynamics, the description of the results of specific statistical analysis is not found. Please supplement the results of statistical analysis.

4. The description of the results is incomplete, please revise it.

6. PLOS authors have the option to publish the peer review history of their article (what does this mean?). If published, this will include your full peer review and any attached files.

Reviewer #1: No

Reviewer #2: No

---

## [Author Response · Author response to Decision Letter 0]

10 May 2022

Journal Requirements:

Response: We have revised our manuscript carefully and we ensure that our manuscript meets PLOS ONE's style requirements, including file naming.

2. Thank you for stating the following financial disclosure: Please state what role the funders took in the study. If the funders had no role, please state: "The funders had no role in study design, data collection and analysis, decision to publish, or preparation of the manuscript." If this statement is not correct you must amend it as needed. Please include this amended Role of Funder statement in your cover letter; we will change the online submission form on your behalf.

Response: This work was supported by Yunnan Fundamental Research Projects [grant no. 202201AT070269]; the National Natural Science Foundation of China [grant no. 31760541]; the Reserve Talent Project of Yunnan’s Young and Middle-aged Academic and Technical Leaders [grant no. 202105AC160071]; the Young Top Talents of “High-level Talents Training Support Program in Yunnan Province” [grant no. YNWRQNBJ2020291]; and the Reserve Talents Project for the 17th Batch of Kunming’s Young and Middle-aged Academic and Technical Leaders [grant no. KMRCH2019023]. One of the corresponding authors, Dr. Xiaoming Zhang, is the funder. And we have stated what role the funder took in the study. Please see lines 476-477 in revised manuscript with track changes.

Response: We have amended our list of authors on the manuscript to ensure that each author is linked to an affiliation. Please see lines 5-9 in revised manuscript with track changes.

4. We note that you have referenced (ie. Bewick et al. [5]) which has currently not yet been accepted for publication. Please remove this from your References and amend this to state in the body of your manuscript: (ie “Bewick et al. [Unpublished]”) as detailed online in our guide for authors

Response: We have reviewed our reference list to ensure that references we have referenced has been accepted for publication.

5. We note that Figure 1 in your submission contain [map/satellite] images which may be copyrighted. All PLOS content is published under the Creative Commons Attribution License (CC BY 4.0), which means that the manuscript, images, and Supporting Information files will be freely available online, and any third party is permitted to access, download, copy, distribute, and use these materials in any way, even commercially, with proper attribution. For these reasons, we cannot publish previously copyrighted maps or satellite images created using proprietary data, such as Google software (Google Maps, Street View, and Earth). We require you to either (1) present written permission from the copyright holder to publish these figures specifically under the CC BY 4.0 license, or (2) remove the figures from your submission:

Response: We agree with the comment. We have removed the Fig. 1. Please see lines 96 and 138 in revised manuscript with track changes.

Response: We have reviewed our reference list to ensure that it is complete and correct.

Reviewers' comments:

Reviewer #1: 

Point 1. The word abondance should be abundance

Response: We agree with the comment. We have revised “abondance” to “abundance”. Please see lines 23, 155, 157, 190, 229 and Table 2 in revised manuscript with track changes.

Point 2. L92: add “each of” before 12 tomato field plots

Response: We agree with the comment. We have added “each of” before “12 tomato field plots”, Please see line 94 in revised manuscript with track changes.

Point 3. L320-321: This sentence is not clear to me

Response: We agree with the comment. We have revised this sentence to make it clearer. Please see lines 424-425 in revised manuscript with track changes.

Point 4. The biodiversity is one key factor influencing the landscape. The authors may have a brief description of the diversity on each of the landscape type. For example, what is the main flower in the flower field landscape? How about other landscape?

Response: We agree with the comment. We have written small sub-paragraph for each of the landscapes describing their diversities. Please see lines 108-129 in revised manuscript with track changes.

Point 5. The predatory natural enemy is always co-occurrence with parasitic natural enemy. How about the role of predatory natural enemies in the field trial sites?

Response: In our study, it was found that the dominant parasitic natural enemy of Bemisia tabaci in different agricultural landscapes was Encarsia formosa, which was the specific natural enemy of whitefly. Similarly, we also investigated predatory natural enemies and found that the main predatory natural enemies were. Nesidiocoris tenuis Reuter (Hemiptera: Miridae), Chrysoperla sinica Tjeder (Neuroptera: Chrysopidae), Menochilus sexmaculata Fabricius (Coleoptera: Coccinellidae) and Harmonia axyridis Pallas (Coleoptera: Coccinellidae). However, most predatory natural enemies are omnivorous, and not only control Bemisia tabaci, due to the abundant data, we are still conducting correlation analysis on the relevant data of predatory natural enemy insects, which will be published later.

Reviewer #2: 

Point 1. The hypothesis is based on finding that the parasitoids of B. tabaci could be observed in the tomato planting fields, but the number of species and black pupae of the wasps were significantly difference under different agricultural landscapes around Kunming, Yunnan Province. However, this is not published data, and it is better to put the content in the manuscript as supplementary material.

Response: We agree with the comment. We have uploaded the data as supplementary data 1 (Data S1). Please see lines 69 in revised manuscript with track changes.

Point 2. Some of the descriptions in the data analysis are confusing. Are the data normally distributed? The data should be transformed if it did not follow a normal distribution. Why use the LSD method instead of Tukey's HSD? 

Response: ① We agree with the comment. We have revised the descriptions in the data analysis to make it clearer. Please see lines 193-194 in revised manuscript with track changes.

② LSD method has wider application scope than Tukey's HSD and it has high inspection efficiency. Tukey's HSD is applicable to the same number of samples in each treatment. Because the sampling number in different landscapes in the same activity period was not always the same in our study, we chose the LSD method.

Point 3. Some parts of the results lack the specific description of the statistical analysis results. For example, the comparison of relative abundance and population dynamics, the description of the results of specific statistical analysis is not found. Please supplement the results of statistical analysis.

Response: ① We have supplemented specific description of the statistical analysis results about the comparison of relative abundance. Please see lines 230-236 in revised manuscript with track changes.

② For population dynamics, we want to show the population growth and decline process of Bemisia tabaci and Encarsia formosa in different agricultural landscapes in this part, so we did not make statistical analysis. Please see lines 281-348 in revised manuscript with track changes.

Point 4. The description of the results is incomplete, please revise it.

Response: We agree with the comment. We have revised the description of the results to make it clearer and more complete. Please see lines 230-236 and 269-348 in revised manuscript with track changes.

---

## [Decision Letter · Decision Letter 1]

18 Jul 2022

Flowering agricultural landscapes enhance parasitoid biological control to Bemisia tabaci on tomato in south China

PONE-D-21-35756R1

Dear Dr. Yang,

We’re pleased to inform you that your manuscript has been judged scientifically suitable for publication and will be formally accepted for publication once it meets all outstanding technical requirements.

Kind regards,

Yonggen Lou

Academic Editor

PLOS ONE

Additional Editor Comments (optional):

Reviewers' comments:

Reviewer's Responses to Questions

**Comments to the Author**

1. If the authors have adequately addressed your comments raised in a previous round of review and you feel that this manuscript is now acceptable for publication, you may indicate that here to bypass the “Comments to the Author” section, enter your conflict of interest statement in the “Confidential to Editor” section, and submit your "Accept" recommendation.

Reviewer #1: All comments have been addressed

Reviewer #2: All comments have been addressed

2. Is the manuscript technically sound, and do the data support the conclusions?

Reviewer #1: Yes

Reviewer #2: Yes

3. Has the statistical analysis been performed appropriately and rigorously? 

Reviewer #1: Yes

Reviewer #2: Yes

4. Have the authors made all data underlying the findings in their manuscript fully available?

Reviewer #1: Yes

Reviewer #2: Yes

5. Is the manuscript presented in an intelligible fashion and written in standard English?

Reviewer #1: Yes

Reviewer #2: Yes

6. Review Comments to the Author

Reviewer #1: (No Response)

Reviewer #2: Accepted. The data have been uploaded as supplementary data, and the manuscrip has been well revised.

7. PLOS authors have the option to publish the peer review history of their article (what does this mean?). If published, this will include your full peer review and any attached files.

Reviewer #1: No

Reviewer #2: No

---

## [Editor Report · Acceptance letter]

22 Jul 2022

PONE-D-21-35756R1 

Flowering agricultural landscapes enhance parasitoid biological control to *Bemisia tabaci* on tomato in south China 

Dear Dr. Yang:

I'm pleased to inform you that your manuscript has been deemed suitable for publication in PLOS ONE. Congratulations! Your manuscript is now with our production department. 

Kind regards, 

on behalf of

Dr. Yonggen Lou 

Academic Editor

PLOS ONE